# Association between biological sex and insecticide-treated net use among household members in ethnic minority and internally displaced populations in eastern Myanmar

**Breagh Cheng[1], Saw Nay Htoo[2], Naw Pue Pue Mhote[3], Colleen M. Davison**[1]*

**1** Department of Public Health Sciences, Queen's University, Kingston, Ontario, Canada, **2** Burma Medical Association, Mae Sot, Thailand, **3** Health Information Systems Working Group, Mae Sot, Thailand

* davisonc@queensu.ca

**Data Availability Statement:** The 2013 EBRMS data used in the present study are collected and owned by the Health Information Systems Working

## Abstract

Malaria prevalence in Myanmar is highest among remote and ethnic minority populations living near forest fringes along the country's international borders. Insecticide-treated nets (ITNs) are a key intervention used to prevent malaria transmission, but insufficient ITN availability and low use can hinder effectiveness. This study assessed age and sex disparities in ITN possession, access, and use among household members of ethnic minority and internally displaced populations in eastern Myanmar. Cross-sectional data from the 2013 Eastern Burma Retrospective Mortality Survey were used to describe prevalence of ITN possession, access, and use. The association between a household member's biological sex and their ITN use was assessed using multilevel log binomial regression. Age and household ITN supply were tested as potential effect modifiers. Of 37927 household members, 89.8% (95% CI: 89.5, 90.1) of people lived in households with at least one ITN. Approximately half belonged to households with sufficient ITN supply and used an ITN. Pregnant women and children under five had the highest proportion of ITN use regardless of sufficient household ITN status. Female adults aged 15 to 49 years old (Risk ratio or RR: 1.4, 95% CI: 1.29, 1.52) were more likely to use ITNs. This relationship did not differ by sufficient household ITN status. The findings suggest that among ethnic minority populations in areas where ITN use is indicated, many households do not have adequate ITN supply, and many individuals are not using ITNs. Children under five and pregnant women appear to be prioritized for ITN use and overall, women are slightly more likely to use ITNs than men. This study's findings can support efforts ensuring that all household members belonging to ethnic minority and displaced populations in Eastern Myanmar benefit from sufficient ITN access and use for malaria prevention.

## Introduction

Myanmar has made significant reductions in malaria mortality and morbidity over the past decade. Between 2010–2018, the number of reported malaria cases in the country declined by

Group in Mae Sot, Thailand. Researchers interested in using this data set must obtain written permission and approval from the local ethics committee and the Health Information Systems Working Group prior to use. Further information about the EBRMS data can be requested from the Health Information Systems Working Group in Mae Sot, Thailand. Data request queries should be sent to ms.ethicsboard@gmail.com. The authors had no special access privileges.

**Funding:** Funding for this study was provided by a Frederick Banting and Charles Best Canada Graduate Scholarship and a Michael Smith Foreign Study Supplement provided through the Canadian Institutes of Health Research (CIHR) and a Research Assistant stipend through a Queen Elizabeth Scholars Advanced Scholars Fund (Davison CM, co-PI). The funder had no role in study design, data collection and analysis, decision to publish, or preparation of the manuscript.

**Competing interests:** The authors have declared that no competing interests exist.

76% and deaths fell by 95% during the same period [1]. Despite these promising reductions, rural residents near forested and mountainous regions still remain at high malaria risk [1–3]. Remote communities in particular bear much of the malaria burden because of increased exposure to malaria vectors and limited access to healthcare services [2,3]. Ethnic and community-based health organizations help address gaps in health care access, however physical inaccessibility and coordination challenges among the mobile health teams continue to hinder effective and timely delivery of essential health services [4].

Long-standing conflict in Myanmar's eastern border states has widened ethnic health disparities. Warfare has weakened public health infrastructure, impeded assistance from the Myanmar government and international humanitarian organizations, and forcibly displaced thousands of residents within Myanmar and across the border to Thailand [5]. Villagers fleeing to the Thailand-Myanmar border led to the formation of six major internally displaced person's (IDP) semi-permanent camps situated primarily in eastern Shan state [6]. These camps, which include Kong Mung Mueang, Loi Kaw Wan, Loi Sam Sip, Loi Lam, Loi Tai Lang, and Ee Tu Hta, held over 8000 IDPs in December 2017 [7,8]. Roughly 2500 people live in each camp with the exception of Loi Sam Sip and Loi Lam, which each have an estimated population of 300 people [9]. As of April 2020, there are 170 other refugee camps and camp-like settlements spread across Kachin and Shan State near the China-Myanmar border [10]. In the same period, the size of occupancy at each camp ranged between 104 to over 34700 people at the largest refugee camp on the Myanmar-Thailand border, Mae La [10]. Accessing health services tends to be even more dire in IDP camps and surrounding formal and informal settlements despite the high prevalence of infectious diseases in these settings due to government restrictions against any international aid and funding, making health outcomes generally worse in these areas [4,11].

Insecticide-treated nets (ITNs) are one of the most effective and available vector control methods to prevent malaria transmission [12,13]. Millions of ITNs and long-lasting insecticide-treated nets (LLINs) have been distributed since 2003 in Myanmar. Distribution has been via community channels with village malaria workers and periodic mass campaigns carried out by Myanmar's National Malaria Control Programme, non-governmental organizations and other donors [14,15]. Since 2011, countries in the Greater Mekong Subregion have transitioned from targeted to universal ITN coverage regardless of household composition [16,17]. To eliminate malaria by the year 2030 and ensure community-wide protection against mosquitoes, Myanmar's National Malaria Strategic Plan (2016 to 2020) aims to achieve universal (one hundred percent) ownership and use of ITNs. Various individual-level factors have been shown to predict ITN use including biological sex [18] and age [19,20]. Most ITN studies, however, have been conducted in African contexts. They generally demonstrate higher ITN use among female household members and younger age categories [21–23]. Comparatively, there have been limited epidemiological studies in Myanmar examining how ITN possession, access, and use varies by different sociodemographic groups previously identified to be at high risk for malaria infection [17–20,24,25]. Children under five and pregnant women are universally considered to be most vulnerable to malaria infection but in Myanmar, school-aged children and adult males aged 15 years and older have recently been suggested to also be at higher risk [18,26]. While some studies have examined ITN possession and use comparing people from the general population or migrant groups [25,27,28], studies have not extensively considered malaria intervention use among ethnic minority and IDP populations [2,6,29] in Myanmar [25,30,31]. Understanding patterns in ITN use among ethnic and displaced populations is important to inform disease prevention efforts as well as ensure equitable coverage and use of available health resources. This study aimed to assess the association between biological sex and ITN use among household members in ethnic minority and IDP populations in eastern

Myanmar. It also aimed to determine whether any association was modified by the age of household members or by the ratio of ITNs available at the household level per person.

## Materials and methods

### Data source and sampling technique

This study used data from the 2013 Eastern Burma Retrospective Mortality Survey (EBRMS). The EBRMS is a population-based, cross-sectional survey and was conducted by the Health Information System Working Group (HISWG) based in Mae Sot, Thailand. The HISWG is a collaboration of ethnic and multi-ethnic community-based organizations in eastern Myanmar that aims to standardize approaches to data collection and analysis and has conducted population surveys since 2002 [8].

The original survey's main objective was to estimate mortality and morbidity in each area serviced by five ethnic and community-based organizations that provide health services to people living in remote and IDP communities in eastern Myanmar.

Multistage cluster sampling was used to select the sample for this survey and was designed to estimate under-five mortality rates in each service area. In the first stage, clusters were chosen using population proportional to size and in the second stage, proximity sampling was used to select 30 households for each cluster. In clusters with less than 30 households, another household was chosen randomly to replace households up to 30. A household was defined as a group of people who live under the same roof for at least two months and who share meals. The sampling frame of 456,786 people (87,841 households) was constructed using village-level population lists provided by ethnic and community-based health organizations that had been updated within the year before the survey. Geographic boundaries of enumeration areas were drawn based on service (or catchment) areas for each health organization [8].

### Data collection

The 2013 EBRMS was administered by the Health Information System Working Group with technical support from member organization representatives and international partners. Data were collected by 80 trained surveyors from July to September 2013 during the rainy period (May to October) where peak malaria transmission occurs and ITN use is expected [2]. Data was collected from village settings and four IDP camps including Kong Mung Moung, Loi Lum, Loi Sam Sip, and Loi Tai Leng, in areas serviced by five ethnic and community-based organizations in Bago, Karen, Karenni, Tanintharyi, Mon, Kachin, Palaung, and Shan states of eastern Myanmar [32].

The 2013 EBRMS collected information on household member demographics, migration, mortality, and women's reproductive health, in addition to general wellness, food security, water and sanitation, healthcare access, exposure to human rights violations and most notably for this study, malaria prevention and treatment. It was written originally in English, translated into three languages (Burmese, Mon, and Sgaw Karen), and translated back into English. The first 78 survey questions were answered by the head of the household (either male or female). If a head of household was unavailable, the woman with the youngest child under five, any currently pregnant woman, and the oldest women of reproductive age in the household at the time of the survey would complete the survey as the designate respondent in descending order of priority. An additional 19 questions on reproductive health were asked of all women of reproductive age in the household who had either a child under five years or were pregnant at the time of the survey. Data were collected from a total of 6620 households. Overall, this survey achieved a 91.5% response rate from household heads [8]. This remains the most recent data available about malaria prevention intervention uptake for this population.

### Dependent variable

The main outcome of interest was the use of an ITN by a household member the night before the survey. This was measured by responses to the survey question "Did this person sleep under an ITN or bed net last night?" [8]. Responses indicating a household member slept under an ITN or bed net were categorized using responses indicating a household member did or did not sleep under an ITN or bed net. The outcome variable was coded as a binary variable with a value of 1 and 0.

### Independent variables

The primary exposure variable was biological sex of a household member. In the 2013 EBRMS, the head of households were asked to indicate the sex (male or female) and age of each member of their household, including themselves. Plausible confounders were selected based on a priori research which included education level and marital status of the household head, as well as ITN supply in the house resided by household members. Age was categorized as <5, 5–14, 15–49, and ≥ 50 years. Marital status was coded as currently married or single (widowed, separated or divorced). Education was coded as none, primary (1–5 standard), secondary (6–10 standard), higher than secondary (Above 10 standard), and other kinds of education (short course or attendance at a monastery). A new variable was created to represent the number of people in a household with the categories 1–3, 4–6, or ≥7 household members. A variable representing ITN supply was created by dividing the number of ITNs owned by each household by the household size. 'Sufficient' ITN supply was defined as an ITN to household member ratio of at least one ITN per every two people and an 'insufficient' supply of ITNs was defined as an ITN to household member ratio less than 1:2 [33]. Any ITN access was defined as household members in households that owned any number of ITNs. Ethnicity was coded as Karen, Karenni, Shan, Mon and other ethnicities (Burmese, other). Respondents who indicated "None" for ethnicity were treated as missing. Household relations were coded as self (the household head or designated [HDD]), parent of HHD, spouse of HHD, child of HHD, and extended relatives and friends of HHD (uncle/aunt, brother/sister, nephew/niece, friend, cousin, other relative). A demographic variable was created with the mutually exclusive categories pregnant females of any age, non-pregnant females aged 15–49 years, adult males aged 15 years and older, and other (representing all other household members). Coding of variables were based on previous works [22,24]. Population level ITN access was calculated based on previous recommendations and related work [34,35]. The number of potential ITN users was calculated by multiplying the number of ITNs in each household by 2.0. In households with more than one net for every two people, the potential ITN users was set equal to the de-facto population (the number of people who slept in the household the night before including visitors [36] in that household if the potential users exceeded the number of people in the household). Population access was calculated by dividing potential ITN users by the number of de-facto members in each household. Potential interactions between age groups and ITN supply and biological sex were tested after obtaining the adjusted regression model.

### Statistical analysis

Analysis excluded individuals with missing data, defined as any response indicating "Don't know," "Missing," "Refused," or "Not available," for key sociodemographic variables including sex, age, ITN supply, and ITN use. ITN possession, access, and use by household members are presented as proportions with their 95% confidence intervals and compared in relation to sociodemographic groups using Rao-Scott chi-square tests. Significance level was set at $P$-value <0.05 for all hypothesis tests.

Multilevel, log binomial regression modelling was used to fit the data using the PROC GLIMMIX procedure with a log link and binomial distribution to generate risk ratios (RR) and their 95% confidence intervals (CI). Data was modelled with biological sex as fixed effects with a random intercept for the effect of household clusters. The intraclass correlation coefficient (ICC) was calculated to measure the extent of potential correlation occurring between individuals in the same households. Backwards elimination was used to build the most parsimonious model after removal of each covariate from the main effects model based on a significance level of p-value (p<0.15). All analyses were conducted using SAS 9.4 (SAS Inc., Cary, North Carolina, USA).

## Ethics statement

The current analysis received ethics clearance and approval from the Community Ethics Advisory Board at the Mae Tao Clinic in Mae Sot, Thailand where the data was collected. It also received ethical review and approval from the Queen's University Health Sciences Research Ethics Board. Ethics approval for the dataset used in this study was also obtained by the Health Information System Working Group from the UCLA Health Ethics Center in Los Angeles, United States. Access to the data was approved after time spent with the partner organization in Thailand and ethical review of the protocol by the local Mae Tao Clinic Ethics Board, Mae Sot, Thailand. Renumeration was not given and informed consent from participants of the original survey were not required for this retrospective analysis.

The 2013 EBRMS data used in the present study are collected and owned by the Health Information Systems Working Group in Mae Sot, Thailand. Researchers interested in using this data set must obtain written permission and approval from the local ethics committee and the Health Information Systems Working Group prior to use. Further information about the EBRMS data can be requested from the Health Information Systems Working Group in Mae Sot, Thailand. Data request queries should be sent to ms.ethicsboard@gmail.com. The authors had no special access privileges.

## Results

### General sample characteristics

Out of 225 sampled clusters, 10 clusters were replaced and six were not accessible leaving a total of 219 clusters with 37,927 household members from 7958 households in the final sample. Households were headed more commonly by males (58.7 95% CI: 57.6, 59.7) compared to females (41.3 95% CI: 40.2, 42.4). The median household size was 4.9 (95% CI: 4.8, 4.9) members. Males and females were roughly equal in proportion. The median age of household members was 21.3 years (95% CI: 21.0, 21.7). Sociodemographic characteristics of respondents from the sampled states and regions in eastern Myanmar are summarized in Table 1.

### ITN possession, use and use conditional on access

After excluding missing values, the median proportion of the *de facto* population with access to an ITN within a household was 85.4% (95% CI: 83.4, 87.3). The proportion of people who belonged to a household owning at least one ITN was 90.4% (95% CI: 90.1, 90.7). The proportion of household members who belonged to a household with a sufficient number of ITNs was 48.4% (95% CI: 47.9, 48.9). Approximately two thirds (68.2%, 95% CI: 67.7, 68.7) of household members slept under an ITN the night before the survey was conducted. The majority (81.6%, 95% CI: 81.0, 82.1) of people belonging to households with sufficient ITN supply also used an ITN. Among people belonging to households with insufficient ITN supply, 55.7%

**Table 1.** Select sociodemographic characteristics of household members in the 2013 eastern Myanmar retrospective mortality survey sample (n = 37927).

| Characteristics | Frequency | Proportion (95% confidence interval) |
|---|---|---|
| Total number of household members | 37927 | |
| Total number of households | 7958 | |
| Male household head or designate (HHD) | 4668 | 58.7 (57.6, 59.7) |
| Female HHD | 3287 | 41.3 (40.2, 42.4) |
| **Biological sex** | | |
| Male | 18684 | 49.3 (48.8, 49.8) |
| Female | 19235 | 50.7 (50.2, 51.2) |
| Missing* | 8 | 0.02 |
| **Age groups** | | |
| < 5 | 4755 | 12.5 (12.2, 12.9) |
| 5–14 | 9237 | 24.4 (23.9, 24.8) |
| 15–49 | 18062 | 47.6 (47.1, 48.1) |
| ≥ 50 | 5688 | 15.0 (14.6, 15.4) |
| Missing | 185 | 0.5 |
| **Other key socio- demographic sub-groups** | | |
| Pregnant females of any age | 635 | 1.7 (1.5, 1.8) |
| Non-pregnant females (of reproductive age 15–49) | 8638 | 22.8 (22.4, 23.2) |
| Adult males ≥ 15 | 11636 | 30.7 (30.2, 31.1) |
| Other | 17018 | 44.9 (44.4, 45.4) |
| Missing | 8 | 0.5 |
| **Highest level of education attended by HHD** | | |
| None | 17430 | 46.0 (45.5, 46.5) |
| Primary | 11850 | 31.2 (30.8, 31.7) |
| Secondary | 4482 | 11.8 (11.5, 12.1) |
| Above secondary | 512 | 1.4 (1.2, 1.5) |
| Other | 3429 | 9.0 (8.7, 9.3) |
| Missing | 224 | 0.57 |
| **Ethnicity of HHD** | | |
| Karen | 17029 | 44.9 (44.4, 45.4) |
| Shan | 7641 | 20.1 (19.7, 20.6) |
| Mon | 6027 | 15.9 (15.5, 16.3) |
| Karenni | 4747 | 12.5 (12.2, 12.8) |
| Other | 1771 | 4.7 (4.5, 4.9) |
| Missing† | 712 | 1.9 |
| **ITN possession** | | |
| Belonged to a household that owned at least 1 ITN | 33059 | 89.8 (89.5, 90.1) |
| Belonged to a household that did not own any ITNs | 3526 | 10.0 (9.7, 10.3) |
| Missing | 81 | 0.21 |
| **Belongs to a household with sufficient access of at least one ITN per two persons** | | |
| Belonged to household with sufficient ITN coverage | 18193 | 48.0 (47.5, 48.5) |
| Belonged to household with insufficient coverage | 19659 | 51.8 (51.3, 52.3) |
| Missing | 75 | 0.20 |
| **ITN use** | | |
| Used ITN the night before | 25342 | 68.2 (67.7, 678.7) |
| Did not use ITN the night before | 12015 | 31.8 (31.3, 32.3) |
| Missing | 570 | 1.5 |
| **Household attributes** | | |

(*Continued*)

**Table 1.** (*Continued*)

| Characteristics | Frequency | Proportion (95% confidence interval) |
|---|---|---|
| Belonged to households with 1–3 people | 5579 | 14.7 (14.4, 15.1) |
| Belonged to households with 4–6 people | 20919 | 55.2 (54.7, 55.7) |
| Belonged to households with 7+ people | 11429 | 30.1 (29.7, 30.6) |
| Missing | 0 | |
| **Household members captured in the survey data** | | |
| HHD | 7964 | 21.0 (20.6, 21.4) |
| Parent of HHD | 1448 | 3.8 (3.6, 4.0) |
| Spouse of HHD | 6382 | 16.8 (16.5, 17.2) |
| Child of HHD | 17973 | 47.4 (46.9, 47.9) |
| Extended relatives and friends of HHD | 4140 | 10.9 (10.6, 11.2) |
| Missing | 20 | 0.05 |

*Missing category defined as 'Don't know,' 'refused,' and 'not applicable' responses.

†Responses reporting no ethnicity were defined as missing for the ethnicity category.

(95% CI: 55.0, 56.4) of household members slept under an ITN the night before the survey. There was a significant difference in ITN use between females and males (p < .001).

A total of 448 pregnant females (73.4%, 95% CI: 69.9, 76.9) and 3363 children under five (72.6%, 95% CI: 71.3, 73.9) used ITNs, which represent the socio-demographic sub-groups with the largest proportion who slept under an ITN the night before the survey. There were 5874 (70.2%, 95% CI: 69.2, 71.2) non-pregnant women of reproductive age in the sample who slept under an ITN the night before the survey, which varies only slightly from ITN use by pregnant women. A total of 7287 (65.4%, 95% CI: 64.5, 66.3) adult males aged 15 years and older slept under an ITN the night before the survey and constituted the group with the lowest proportion of ITN use.

Overall, there was a significant difference in ITN use based on whether or not a household had sufficient ITN supply (p-value < .001). For people in households with sufficient ITN supply, the proportion of ITN use was slightly higher for all sub-groups as compared to use among people in households without adequate supply. When considering sufficient ITN supply status of a household, a total of 1541 children under five used an ITN (85.2%, 95% CI: 83.6, 86.8), who remained the socio-demographic sub-group with the highest proportion of ITN use. The next highest socio-demographic sub-groups were pregnant and non-pregnant females; a total of 257 pregnant females of any age (82.6, 95% CI: 78.4, 86.8) and 4290 non-pregnant females between 15–49 years of age (82.8%, 95% CI: 81.7, 84.0) who belonged to households with sufficient ITN supply slept under an ITN the night before the survey. A total of 4587 adult males aged 15 years and older (79.7, 95% CI: 78.7, 80.7) slept under an ITN the night before, which remained the sub-group with the lowest proportion of ITN use (Table 2).

**Measures of variation.** Studies using complex survey designs may result in correlated data, which occurs when observations within one cluster may be more similar to each other than they are to observations from other clusters. This type of data may not meet assumptions of independence [37]. In the null model (i.e. without explanatory factors added), the household-level ICC was 0.001. This indicates that only 0.1% (at the household level) of the overall variance is accounted for by clustering (correlation) of data points at these levels. This is considered a low ICC value meaning that data from the same household or household cluster are not showing high levels of clustering/correlation.

**Table 2. Proportion of ITN use (sleeping under ITNs the prior night) and non-use among each socio-demographic sub-group who belonged to households with any and sufficient ITN access in eastern Myanmar, 2013 (n = 36585).**

| | Lives in household with any ITN access (n = 36585) | | | Lives in household with sufficient ITN access (n = 17694) | | |
|---|---|---|---|---|---|---|
| | Used ITN (n = 24954) | | | | Used ITN (n = 14430) | |
| **Groups** | n (%) | 95% CI | p-value | **Groups** | n (%) | 95% CI | p-value |
| **Sex** | | | < .001* | | | | 0.02 |
| Males (n = 18003) | 12067 (67.0) | 66.4, 67.7 | | Males (n = 8522) | 6891 (80.9) | 80.0, 81.7 | |
| Females (n = 18582) | 12887 (69.4) | 68.7, 70.0 | | Females (n = 9172) | 7539 (82.2) | 81.4, 83.0 | |
| **Age groups** | | | < .001 | | | | < .001 |
| < 5 (n = 4631) | 3363 (72.6) | 71.3, 73.9 | | < 5 (n = 1809) | 1541 (85.2) | 836, 86.8 | |
| 5–14 (n = 8993) | 6103 (67.9) | 66.9, 68.8 | | 5–14 (n = 3782) | 3121 (82.5) | 81.3, 83.7 | |
| 15–49 (n = 17473) | 11947 (68.4) | 67.7, 69.1 | | 15–49 (n = 8849) | 7248 (81.9) | 81.1, 82.7 | |
| ≥ 50 (n = 5488) | 3541 (64.5) | 63.3, 65.8 | | ≥ 50 (n = 3254) | 2520 (77.4) | 76.0, 78.9 | |
| **Socio- demographic sub-groups** | | | < .001 | | | | < .001 |
| Pregnant females of any age (n = 610) | 448 (73.4) | 69.9, 76.9 | | Pregnant females of any age (n = 311) | 257 (82.6) | 78.4, 86.8 | |
| Non-pregnant females 15–49 (n = 8379) | 574 (70.1) | 69.1, 71.1 | | Non-pregnant females 15–49 (n = 4290) | 3554 (82.8) | 81.7, 84.0 | |
| Adult males ≥ 15 (n = 11141) | 7287 (65.4) | 64.5, 66.3 | | Adult males ≥ 15 (n = 5756) | 4587 (79.7) | 78.7, 80.7 | |
| Other (16455) | 11345 (68.9) | 68.3, 69.6 | | Other (7337) | 6032 (82.2) | 81.3, 83.1 | |

*Post-hoc corrections were not conducted for multiple comparison.

**Multivariable analysis: Relationship between biological sex and ITN use.** There was evidence of a significant interaction between biological sex and age group of a household member but not by ITN supply (p = 0.3), demonstrating that the relationship between biological sex and ITN use varies by age group. After adjusting for confounders, it appears there was a significant association between female sex and ITN use for adults who are aged 15–49 years (RR: 1.4, 95% CI: 1.30, 1.54, p-value < .001). Female sex did not show evidence of a significant association with ITN use for children under five, those aged 5–14 years, or those aged 50 and older (Table 3).

**Table 3. Crude and adjusted risk ratios of the association between biological sex and ITN use by household members in eastern Myanmar stratified by age groups, 2013 (n = 36585).**

| Variable | Unadjusted Model RR (95% CI) | P-value | | Adjusted Model RR (95% CI) | P-value |
|---|---|---|---|---|---|
| **Biological Sex** | | **0–5 years (n = 4631)** | | | |
| Male | 1.00 (ref) | | | 1.00 (ref) | |
| Female | 0.982 (0.980, 1.2) | 0.9 | | 0.958 (0.80, 1.15) | 0.6 |
| | | **5–14 years (n = 8993)** | | | |
| Male | 1.00 (ref) | | | 1.00 (ref) | |
| Female | 1.03 (0.91, 1.16) | 0.7 | | 1.03 (0.91, 1.16) | 0.7 |
| | | **15–49 years (n = 17473)** | | | |
| Male | 1.00 (ref) | | | 1.00 (ref) | |
| Female | 1.4 (1.29, 1.52) | < .001 | | 1.4 (1.30, 1.54) | < .001 |
| | | **50 > years (n = 5488)** | | | |
| Male | 1.00 (ref) | | | 1.00 (ref) | |
| Female | 1.13 (0.98, 1.32) | 0.1 | | 1.17 (1.00, 1.37) | 0.06 |

*P-value for interaction between sex and age group was significant at P < .001.

*P-value for interaction between sex and ITN supply was not significant at P = 0.3.

†Model adjusted for ITN supply, education level and marital status of household head.

## Discussion

In this study, we found that while most households had at least one ITN, only 48.4% of people lived in households with sufficient ITNs in the villages and IDP camps represented in the Eastern Burma Retrospective Mortality Survey in 2013. Approximately two thirds of household members used ITNs the night before the survey, which was even higher among people in households with sufficient ITN supply. Similarly high ITN ownership levels have been reported in previous household assessments among village and IDP populations across Myanmar [24,25,28].

Pregnant women and children had the highest proportion of ITN use, with an even higher proportion of ITN use when these groups belonged to households with an adequate ITN to household member ratio. This indicates that at-risk groups for malaria may be prioritized for ITN use within the household and suggests that having adequate access to ITNs within households may influence the degree of ITN use. Previous studies in Myanmar and across several African countries have reported similarly high rates of ITN use among pregnant females and children under five in villages and IDP camp populations, with even higher rates of ITN use in those belonging to households with sufficient ITN supply compared to those that did not [16,18].

Despite pregnant women and children under five having the highest rates of ITN use compared to others, a significant proportion of other household members remain who do not have access to sufficient ITN supply and do not use ITNs. This is especially important given that the incidence of *Plasmodium falciparum* malaria can vary depending on factors in addition to gender, including occupation type and residence in a village or IDP camp [29]. The patterns of ITN use found in this study differ by sex and age groups who are suggested to be also at risk for malaria in Myanmar, namely adult males older than age 15 years. This population subgroup is generally over-represented among reported malaria cases and at higher risk of clinical falciparum malaria infections [18,32,38]. Expanded support for community-based health organizations that provide healthcare services to those in isolated villages and IDP camp settings who otherwise lack access to medical care is one possible approach that may help ensure that ITNs are made available for all household members [29].

Several reasons may explain the weak yet significant association between female sex and use of ITNs, as well as the overall patterns of ITN use. The lower proportion of ITN use that was identified in this study among adult males may reflect lower perceived biological vulnerability to malaria that contributes to a lack of motivation for, or limited prioritization of, ITN use. This interpretation is supported by a qualitative study on malaria prevention behaviour of male youth living along the Cambodia-Vietnam border [39]. ITN distribution campaigns conducted near the time of the survey may have also been able to achieve close to equal ITN distribution or access between males and females [22]. Village health workers have likely contributed to behavioural changes so that there is prioritization of groups at highest risk for malaria in eastern Myanmar [18,40,41]. Future qualitative research may lend additional understanding about non-ITN use especially when ITNs are available.

ITN ownership alone is often not sufficient for protection against malaria and sociodemographic factors have been widely examined to understand reasons influencing ITN use for malaria prevention, including age and socioeconomic status [22,42]. Beyond these demographic predicators, other factors contributing to ITN use, such as malaria-related knowledge, have been explored with recognition of the complex and diverse reasons for ITN use patterns in Myanmar and other countries [14]. Evidence from qualitative research suggests individual attitudes and social factors such as forest-related occupations (i.e. inconveniences of ITN use) are additional key reasons that influence ITN use by migrant and mobile populations [20,43].

Data on ITN access and use is generally limited in areas controlled by non-state actors that are not accessible by the National Malaria Control Programme [44]. This study's findings on the use of ITNs for malaria prevention by displaced populations living in remote and ethnic states may contribute to a more coherent understanding of ITN use and malaria prevention over time in areas where there has been limited other available evidence (8). Given aims to ensure universal ITN coverage of at-risk populations including ethnic minority groups, findings from this study contribute additional information on malaria intervention use from diverse populations and geographic areas and help examine trends in traditionally under-served areas of Myanmar [14,45].

To address the limited evidence about specific influences on ITN usage in ethnic minority and displaced persons contexts in Myanmar, future research could investigate the influences, such as perceived malaria vulnerability, on rates of ITN utilization among household members in different cultural and geographic contexts in Myanmar [20]. It would be helpful for future studies to examine whether ITN use differences between demographic groups are related to perceived higher vulnerability of pregnant women and children under five to malaria by household members, or perhaps other things. Further qualitative study in this area could further clarify findings.

The strengths of this study include contributing novel information on sex differences in ITN use and how ITN use varies across age groups in both village and IDP camp populations, who remain generally understudied in Myanmar. The study has a high response rate (91.5%) and minimizes the potential for overestimated standard error of estimates by accounting for potential clustering in the data through the use of a fixed effects regression model [46,47].

A key limitation is that the results of the 2013 data may not represent the present malaria situation in eastern border regions of Myanmar today. Although more recent malaria-specific data equivalent to the EBRMS do not yet exist, the malaria landscape in eastern Myanmar has changed somewhat since the data for this study were collected. Since 2012, reported national malaria cases have declined and the country has made commitments to eliminate malaria by 2030 [44,48]. Unfortunately, prevention efforts have been limited in a context of conflict and civil unrest across the country including most recently in response to a military coup [8,37,48] What likely have not changed are the gender- or age-specific cultural norms that seem to underpin some of the ITN usage patterns we found in this study. The 2013 EBRMS data still provide a rare window into population health in ethnic and minority states in Eastern Burma. The results of this study contribute to important gaps in understanding malaria ITN use in this context and remain valuable in the continued efforts for health promotion and prevention in these regions. Other limitations of this study include the potential for unmeasured confounding as information on perceptions of malaria susceptibility and knowledge, were unavailable and thus could not be adjusted for in the model. Furthermore, self-reported data were used to measure ITN use in this study. There is the possibility for potential social desirability bias to create differential misclassification of the outcome and result in spuriously high ITN use estimates [49]. Post-hoc corrections were not conducted for multiple comparisons in this study due to the small number of planned comparisons although it is possible this increases the likelihood of a type I error in our findings [50].

## Conclusion

This study found that female household members between the ages of 15–49 years were more likely to sleep under an ITN the night prior to the survey as compared to males in the same age group. This association was modified by age group of household members but was not modified by ITN supply. Although there was high ITN ownership, sufficient ITN supply and use

differed by age groups among remote and ethnic minority populations. Findings can be used to inform distribution strategies and promote use of ITNs among all household members belonging to ethnic minority and displaced populations in Eastern Myanmar.

## Acknowledgments

We thank the Health Information System Working Group and the Burma Medical Association in Mae Sot, Thailand for their collaboration and access to the Eastern Burma Retrospective Mortality Survey data. We also acknowledge the Queen's University faculty members who provided valuable input to improve the quality of manuscript.

## Author Contributions

**Conceptualization:** Breagh Cheng, Naw Pue Pue Mhote, Colleen M. Davison.

**Data curation:** Breagh Cheng, Naw Pue Pue Mhote.

**Formal analysis:** Breagh Cheng.

**Methodology:** Breagh Cheng, Naw Pue Pue Mhote.

**Project administration:** Saw Nay Htoo, Naw Pue Pue Mhote, Colleen M. Davison.

**Resources:** Saw Nay Htoo, Naw Pue Pue Mhote.

**Supervision:** Saw Nay Htoo, Naw Pue Pue Mhote, Colleen M. Davison.

**Writing – original draft:** Breagh Cheng.

**Writing – review & editing:** Saw Nay Htoo, Naw Pue Pue Mhote, Colleen M. Davison.

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
