## [Decision Letter · Decision Letter 0]

15 Feb 2021

PONE-D-20-40284

Association between biological sex and insecticide-treated net use among household members in ethnic minority and internally displaced populations in eastern Myanmar

PLOS ONE

Dear Dr. Davison,

Thank you for submitting your manuscript to PLOS ONE. After careful consideration, we feel that it has merit but does not fully meet PLOS ONE’s publication criteria as it currently stands. Therefore, we invite you to submit a revised version of the manuscript that addresses the points raised during the review process. The paper needs more careful review of statistical analyses and justification around the use of data from 2013.

We look forward to receiving your revised manuscript.

Kind regards,

Bidhubhusan Mahapatra, Ph.D.

Academic Editor

PLOS ONE

Journal Requirements:

2. In statistical methods, please refer to any post-hoc corrections to correct for multiple comparisons during your statistical analyses. If these were not performed please justify the reasons. Please refer to our statistical reporting guidelines for assistance (https://journals.plos.org/plosone/s/submission-guidelines.#loc-statistical-reporting).

3. In your statistical analyses, please state whether you accounted for clustering by locality. For example, did you consider using multilevel models?

4.We note that you have indicated that data from this study are available upon request. PLOS only allows data to be available upon request if there are legal or ethical restrictions on sharing data publicly. For more information on unacceptable data access restrictions, please see http://journals.plos.org/plosone/s/data-availability#loc-unacceptable-data-access-restrictions.

5. We note you have included a table to which you do not refer in the text of your manuscript. Please ensure that you refer to Table 0 in your text; if accepted, production will need this reference to link the reader to the Table.

6. Please include a copy of Table 3 which you refer to in your text on page 12.

Additional Editor Comments:

This is an interesting area of work. While there is value in this work, there are certain areas where the authors need to work to strengthen the paper. First, the paper needs a strong justification on why you are using data from 2013? Given the development that has taken place in Myanmar in last few years, does this data still hold significance? Please include justification around it. Second, the statistical analyses need a thorough review. I have noticed several errors across the tables. to give a few example: the 95% CIs are not consistent with the point estimates in Table 1 for age 50+ (did not use ITN). Also, in Table 2, Lives in household with sufficient ITN supply for comparison by sex of HH member, the 95% Cis overlap, but p-value is 0.03 which does not look correct. Some of other minor observations are:(i) How the missing values were treated in the analysis? (ii) Multivariable analysis instead of multivariate analysis, (iii) give more clarity on the sampling of households, (iv) why local IRB approval was not taken for this study? (v) Who was responsible for collecting data and when exactly data was collected, give clarity around it? Clarify if the duration of data collection had any effect on the ITN use or other survey measures? (vi) Rename Table 0 as Table 3? (vii) Include interaction analysis result in the table, (viii) In the discussion line 271, you suggest that ITN use was self-reported. Does it mean all HH members were asked questions oon ITN use? If not, then change it accordingly. (ix) A key limitation is the use of a data that may not be representing the current situation. Suggest including this as a limitation.

Reviewers' comments:

Reviewer's Responses to Questions

**Comments to the Author**

1. Is the manuscript technically sound, and do the data support the conclusions?

Reviewer #1: Partly

Reviewer #2: Yes

Reviewer #3: Partly

2. Has the statistical analysis been performed appropriately and rigorously? 

Reviewer #1: Yes

Reviewer #2: Yes

Reviewer #3: Yes

3. Have the authors made all data underlying the findings in their manuscript fully available?

Reviewer #1: Yes

Reviewer #2: Yes

Reviewer #3: Yes

4. Is the manuscript presented in an intelligible fashion and written in standard English?

Reviewer #1: Yes

Reviewer #2: Yes

Reviewer #3: Yes

5. Review Comments to the Author

Reviewer #1: The authors explored ITN access and use among a high risk migrant population in Myanmar using data from 2013. They demonstrate differences in ITN use by age and biologic sex and that pregnant women and children under five are prioritized to use ITNs. Their findings may potentially be used inform the distribution strategies and promote use of ITNs among this vulnerable population.

Major

The manuscript will hugely benefit from a detailed justification of their use of data from 2013, particularly given the change in malaria landscape since 2013. The authors note that Between 2010-2018, the number of reported malaria cases in the country declined by 76% and deaths fell by 95%. The context of the migrant population in Myanmar has also changed with the Rohingya crises and over one million people in need of humanitarian assistance. This manuscript would be bolstered significantly by the use of more recent data or a clear rationale for the use of such dated data, delineating how the study findings are still relevant in 2021. It would also be helpful to note any differences in the study population, ITN distribution channels and major factors influencing ITN access and use have changed from the time of the study to date. Of note, the study findings probably do not reflect the more recent shift to universal coverage of ITN ownership and use which may have occurred since 2013.

Minor

1. Overall

Current thinking as evidenced by the recent literature and global research such as Malaria Indicator Surveys, Demographic and Health Surveys and Multiple Indicator Cluster Surveys now highlight population level access to ITNs. I would encourage the authors include overall population level ITN access in their findings. Of note, while this indicator does not allow exploration of individual level factors, it might benefit the manuscript to show how the population level ITN access and potential differences by household or regional level factors. For more information on the methodology, see https://journals.plos.org/plosone/article?id=10.1371/journal.pone.0097496: First, an intermediate variable of “potential ITN users” was created by multiplying the number of ITN in each household by a factor of 2.0. In order to adjust for households with more than one net for every two people, the potential ITN users were set equal to the de-facto population in that household if the potential users exceeded the number of people in the household. Second, the population access indicator was calculated by dividing the potential ITN users by the number of de-facto members for each household and determining the overall sample mean of that fraction.

2.Abstract and Introduction

Line 77-78. Consider revising the sentence: A key challenge to this goal is the effective use of ITNs. It reads as if the major challenge of the goal of using ITNs is the use of ITNs which is probably not what the authors intended.

3. Table 1

Some of the columns do not add up to 100%. I would recommend the authors make the characteristic- Other socio- demographic sub-groups- mutually exclusive by including an "other" category and revising the characteristic to "socio- demographic sub-groups".

Also, the final characteristic- household members- should be revised in order to add up to 100%.

4. Table 2

Include the p values for each of the other socio-demographic sub-groups

5. Discussion

As noted earlier, please discuss the dated findings in light of potential changes to the study population context, ITN distribution strategies, and factors influencing ITN access and use.

6. Conclusion

This is notably absent from the main text. Kindly include

Reviewer #2: This was a well-written paper examining a question that is of interest currently - namely the role of gender on ITN use. Also, Myanmar is a country about which little is known on the subject of malaria prevention. This paper provides some interesting insights.

Reviewer #3: 1. The abstract section needs to be revised and it should be as per PLOS One style and format

2. In Materials and methods section, in page 6: the authors can add Measures section before the variables used and dependent variable/outcome variable sub section and independent variable/explanatory variable section (effect modifiers and confounding variables) section should be created.

3. Again, it will be good if the authors can write detail description on the variables used in the analysis (like categories, definition etc.)

4. The current form of results and discussion section looks weak. If the additional analysis on prevalence of malaria in the region/geography can be added (for sex and geography), along with trend, pattern of malaria prevalence/incidence it will be good. Additionally, the association between ITN use and malaria prevalence/incidence will be very good for future policy and program enhancement and take away.

5. Without the revised results and analysis, the paper discussion section feels like half knowledge has been provided in the region. The discussion section needs to be strengthen with the revised analysis and findings. Then possibly the recommendations and conclusion can be better placed.

6. PLOS authors have the option to publish the peer review history of their article (what does this mean?). If published, this will include your full peer review and any attached files.

Reviewer #1: No

Reviewer #2: No

Reviewer #3: No

---

## [Author Response · Author response to Decision Letter 0]

27 Apr 2021

We have attached a detailed response to reviewers chart.

---

## [Decision Letter · Decision Letter 1]

25 May 2021

Association between biological sex and insecticide-treated net use among household members in ethnic minority and internally displaced populations in eastern Myanmar

PONE-D-20-40284R1

Dear Dr. Davison,

We’re pleased to inform you that your manuscript has been judged scientifically suitable for publication and will be formally accepted for publication once it meets all outstanding technical requirements.

Kind regards,

Bidhubhusan Mahapatra, Ph.D.

Academic Editor

PLOS ONE

Additional Editor Comments (optional):

Reviewers' comments:

Reviewer's Responses to Questions

**Comments to the Author**

1. If the authors have adequately addressed your comments raised in a previous round of review and you feel that this manuscript is now acceptable for publication, you may indicate that here to bypass the “Comments to the Author” section, enter your conflict of interest statement in the “Confidential to Editor” section, and submit your "Accept" recommendation.

Reviewer #1: All comments have been addressed

Reviewer #2: All comments have been addressed

2. Is the manuscript technically sound, and do the data support the conclusions?

Reviewer #1: Yes

Reviewer #2: Yes

3. Has the statistical analysis been performed appropriately and rigorously? 

Reviewer #1: Yes

Reviewer #2: Yes

4. Have the authors made all data underlying the findings in their manuscript fully available?

Reviewer #1: Yes

Reviewer #2: No

5. Is the manuscript presented in an intelligible fashion and written in standard English?

Reviewer #1: Yes

Reviewer #2: Yes

6. Review Comments to the Author

Reviewer #1: (No Response)

Reviewer #2: (No Response)

7. PLOS authors have the option to publish the peer review history of their article (what does this mean?). If published, this will include your full peer review and any attached files.

Reviewer #1: No

Reviewer #2: No

---

## [Editor Report · Acceptance letter]

9 Jun 2021

PONE-D-20-40284R1 

Association between biological sex and insecticide-treated net use among household members in ethnic minority and internally displaced populations in eastern Myanmar 

Dear Dr. Davison:

I'm pleased to inform you that your manuscript has been deemed suitable for publication in PLOS ONE. Congratulations! Your manuscript is now with our production department. 

Kind regards, 

on behalf of

Dr. Bidhubhusan Mahapatra 

Academic Editor

PLOS ONE